# Extremely Rare Case of Fetal Anemia Due to Mitochondrial Disease Managed with Intrauterine Transfusion

**DOI:** 10.3390/medicina58030328

**Published:** 2022-02-22

**Authors:** Jinha Chung, Mi-Young Lee, Jin-Hoon Chung, Hye-Sung Won

**Affiliations:** Department of Obstetrics and Gynecology, Asan Medical Center, University of Ulsan College of Medicine, 88, Olympic-ro 43-gil, Songpa-gu, Seoul 05505, Korea; waterlilyv@naver.com (J.C.); poptwinkle@hanmail.net (M.-Y.L.); sabi0515@hanmail.net (J.-H.C.)

**Keywords:** anemia, blood transfusion, case report, hydrops fetalis, mitochondrial diseases, Pearson syndrome

## Abstract

This report describes a rare case of fetal anemia, confirmed as a mitochondrial disease after birth, treated with intrauterine transfusion (IUT). Although mitochondrial diseases have been described in newborns, research on their prenatal features is lacking. A patient was referred to our institution at 32 gestational weeks owing to fetal hydrops. Fetal anemia was confirmed by cordocentesis. After IUT had been performed three times, the anemia and associated fetal hydrops showed improvement. However, after birth, the neonate had recurrent pancytopenia and lactic acidosis. He was eventually diagnosed with Pearson syndrome and died 2 months after birth. This is the first case report of fetal anemia associated with mitochondrial disease managed with IUT.

## 1. Introduction

Fetal anemia is a rare condition that can cause fetal hydrops, fetal distress, and eventually fetal death if left untreated. Fetal anemia has many causes. However, the leading cause is an alloimmune disease, and the incidence has been rapidly decreasing owing to the use of immunoglobulins [1]. Nonimmune-related causes include parvovirus B19 infection, fetomaternal hemorrhage, monochorionic pregnancy complications, and other rare diseases [2]. Intrauterine transfusion (IUT) effectively treats fetal alloimmune anemia and nonimmune-related diseases [3]. However, to date, few reports have been published regarding the outcomes of rare diseases that cause fetal anemia. Herein, we report an extremely rare case of fetal anemia associated with Pearson syndrome. Although there was a limit to prenatal diagnosis, this rare mitochondrial disease can cause fetal anemia [4]. This is the first report on fetal anemia caused by mitochondrial disease managed with IUT.

## 2. Case Presentation

A 38-year-old, high-risk pregnant primiparous woman visited our hospital at 32 gestational weeks owing to fetal hydrops and oligohydramnios observed on ultrasonography. She had no remarkable medical or family history. Antenatal examination conducted at another hospital 3 weeks before her first visit to our hospital revealed no abnormal findings. The fetus was hydropic with generalized skin edema, ascites, pericardial effusion, cardiomegaly, and bilateral hydrocele (Figure 1A,B). The estimated fetal weight was 1873 g (50th percentile), which might have been overestimated owing to fetal ascites. No other structural abnormalities were noted.

Doppler ultrasonography revealed that the middle cerebral artery peak systolic velocity (MCA-PSV) was 87 cm/s (1.95 MoM) [5], indicating fetal anemia (Figure 1C). Cordocentesis confirmed severe anemia with a hemoglobin level of 1.4 g/dL. To prevent abrupt changes in the fetal cardiovascular system and to improve fetal anemia, IUT was performed three times at 32.0, 32.1, and 32.4 gestational weeks with transfused volumes of 34, 50, and 48 cm^3^, respectively. After the IUTs, the MCA-PSV and hemoglobin levels improved to 56 cm/s (1.3 MoM) and 9.8 g/dL, respectively. The toxoplasmosis, rubella, cytomegalovirus, and herpes simplex virus (TORCH) panel test and parvovirus B19 test from the maternal blood and amniotic fluid were negative. Fetal karyotyping from the amniotic fluid revealed normal findings. Maternal ABO/Rh typing and antibody screening tests revealed O, Rh+ and negative antibodies to rule out alloimmune fetal anemia. In addition, the blood type of the fetus examined with cordocentesis was O, Rh+.

The patient was followed up weekly. The fetal hydropic features, generalized skin edema, ascites, pericardial effusion, cardiomegaly, and bilateral hydrocele were fully resolved two weeks after IUT, and the umbilical artery and ductus venosus Doppler ultrasonography findings were normal. The MCA-PSV was 54.19 cm/s (1–1.3 MoM); thus, no additional IUT was required. The patient underwent elective cesarean delivery at 39.0 gestational weeks and gave birth to a 2110 g male neonate, with 1 and 5 min Apgar scores of 6 and 7, respectively. The neonate’s birth weight was below the third percentile. No gross anomalies were noted.

The neonate was admitted to the neonatal intensive care unit for further evaluation. Initial abdomen ultrasonography showed diffuse dense calcifications in the bilateral adrenal glands. Brain ultrasonography and echocardiography revealed no structural abnormalities. The neonate’s blood type was O, Rh+, and antibody screening was negative. In addition, the TORCH study and direct Coombs’ test were negative. However, initial white blood cell (WBC), hemoglobin (Hb), and platelet (Plt) levels were 3400/μL, 6.5 g/dL, and 156,000/μL, respectively. The initial lactic acid level was 8.0 mmol/L. Peripheral blood smear revealed sideroblastic anemia. Laboratory examinations responded only temporarily to blood transfusion and revealed recurrent pancytopenia and lactic acidosis within a few days. The patient was discharged after repeated blood transfusions. When the patient was rehospitalized due to pancytopenia, his WBC, Hb, and Plt levels were 2500/μL, 6.1 g/dL, and 25,000/μL, respectively. Wolman disease or mitochondrial disease was most likely. His triglycerides, cholesterol, and cortisol levels were normal; thus, the likelihood of Wolman disease was low. Whole exome sequencing for lysosomal acid lipase enzyme gene, mitochondrial deoxyribonucleic acid (DNA) gene mutation, urine organic acid analysis, and acylcarnitine profiling were performed on the 10th day after birth. Mitochondrial gene mutation (m.9424_14840del of the mitochondrial genome) was observed by polymerase chain reaction sequencing using peripheral blood with normal findings of the other studies. The neonate was diagnosed with Pearson syndrome, a mitochondrial disease. Having pancytopenia, the neonate was prone to infection. He died of sepsis secondary to a hospital-acquired *Klebsiella pneumoniae* infection at two months.

## 3. Discussion

Owing to the advances in diagnosis and treatment in the last 20 years, the survival rate of patients with fetal anemia has improved. For decades, fetal anemia had been diagnosed using invasive techniques such as amniocentesis and cordocentesis. In 2000, MCA-PSV Doppler assessment was introduced to predict fetal anemia. However, it has a 10–18% false positivity rate, requiring cordocentesis for confirmation [6]. Nonetheless, it is still widely used as a noninvasive technique for screening fetal anemia [5].

Known causes of fetal anemia include alloimmune anemia, congenital infection, hemorrhage, monochorionic pregnancy complications, and fetal or placental tumors [2,7]. However, only a few reports have described rare causes of fetal anemia. For example, Amann et al. [7] reported 15 cases of fetal anemia, the causes of which were determined after birth. Among the causes were Blackfan–Diamond anemia, elliptocytosis, hemochromatosis, and mucopolysaccharidosis type VII.

Mitochondrial diseases are rare, with a prevalence of 1 in 10,000 births [8], and can be caused by mitochondrial or DNA mutations. The incidence of mitochondrial DNA (mtDNA) deletion disorder does not increase with maternal age [9]. These diseases present a complexity of clinical features and genetic factors, and various features can be exhibited in different organs. Before 4 years of age, the fatality rate is reportedly 57% [10]. The presented case is the first report on a fetus with mitochondrial disease managed with IUT. The fetus showed antenatal features of mitochondrial disease such as fetal growth restriction, oligohydramnios, and fetal anemia [4].

Pearson syndrome, a mitochondrial disease, occurs due to a single, large-scale mtDNA deletion gene defect. The main clinical features of Pearson syndrome are sideroblastic anemia, lactic acidosis, and pancreatic exocrine disorder [11]. Sideroblastic anemia is transfusion-dependent anemia, which might explain why IUT resolved the fetal anemia-associated hydrops in this case. However, studies on the perinatal outcomes of Pearson syndrome with severe fetal anemia are lacking.

## 4. Conclusions

Although extremely rare, mitochondrial disease may cause severe fetal anemia, which can be treated with IUT. However, congenital genetic disorders or hematologic diseases might have a worse prognosis after birth. Therefore, the findings in this report are meaningful in counseling parents about the need for postnatal evaluation and the possibility of fatal prognosis, depending on the diagnosed or suspected cause of fetal anemia.

## Figures and Tables

**Figure 1 medicina-58-00328-f001:**
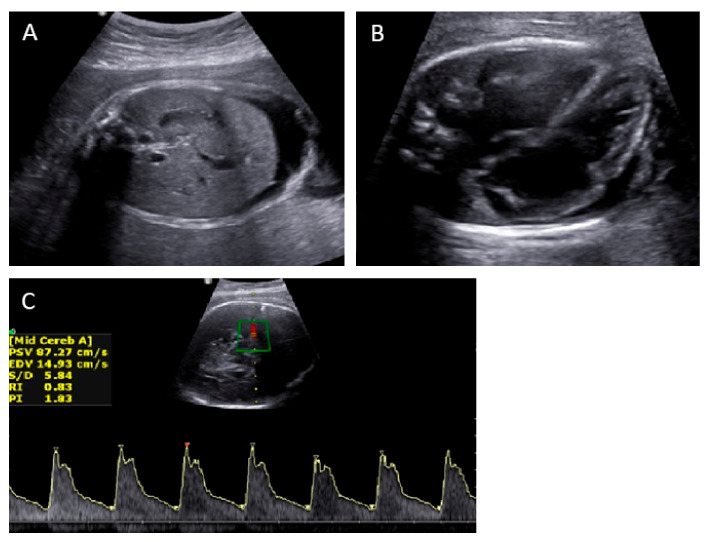
Initial ultrasonographic findings show fetal ascites (**A**) and pericardial effusion with cardiomegaly (**B**). Middle cerebral artery peak systolic velocity on Doppler ultrasonography before intrauterine transfusion was 87 cm/s (1.95 MoM) (**C**).

## Data Availability

The data are available upon request.

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
