# Peer review of "Extremely Rare Case of Fetal Anemia Due to Mitochondrial Disease Managed with Intrauterine Transfusion"

_medicina, 2022, doi:10.3390/medicina58030328_

Round 1

Reviewer 1 Report

I read an extremely interesting manuscript written by Jinha Chung et al.

There are few aspects that I consider imperative to be changed. I will present them in order of appearance in the text. The manuscript needs more data to be presented in order to be valuable as reference point for others specialists.

  1. line 15 – “to the best of my knowledge” - the article has 3 authors. Who wrote this sentence?
  2. I think that it would be more appropriate if the abstract had several subsections, like Introduction, Case presentation, Conclusion.
  3. lines 24-26 – please rewrite. Using “;” seems unfitted here.
  4. lines 31-34 – As presented here, some may assume that you have diagnosed the Pears syndrome during pregnancy. Please rewrite.
  5. lines 37-38 – yes, but she presented a high risk pregnancy, being a 38-year-old primiparous. This is something that needs to be emphasized.
  6. line 52 – “fetal hydrops” all features presented at line 40-41 were solved?
  7. line 53 – “Doppler ultrasonography findings remained normal” – ultrasound of which are? Also, present some values for the middle cerebral artery peak systolic velocity.
  8. line 59 – “no other structural abnormalities” - compared to what? Or apart from what?
  9. line 60 – being a case report, you should give more information about the patient you presented. You should offer some lab values to define pancytopenia and lactic acidosis. How severe were they? Was the severity high from the beginning, or did it get progressively worse?
  10. line 62 – at what age were the genetic tests performed?
  11. line 66 – it appears that everything was under control, the prognosis was good, and he died due to a hospital-acquired infection….. Perhaps you could rephrase this sentence. Having (severe or not) pancytopenia, the child was prone to severe infections
  12. lines 75-78 versus line 79 - MCA-76 PSV Doppler assessment is therefore only a screening method? Does Doppler ultrasonography of the MCA has some detection limit for fetal anemia concerning the systolic velocity?
  13. line 78 - Does any fetus with anemia need amniocentesis?
  14. lines 87-89 - Because of the mother's age, I think it's important to point out that incidence of mtDNA deletion disorders does not increase with maternal age.
  15. line 91-92 - Where did you specify these features? (line 40-41 – “generalized skin edema, ascites, pericardial effusion, cardiomegaly, and bilateral hydrocele”)
  16. line 95 – please explain “mtDNA”
  17. lines 97-98 – “the fetal anemia-associated hydrops was resolved by IUT in this case” – but in lines 39-40 you presented the following “The fetus was hydropic with generalized skin edema, ascites, pericardial effusion, cardiomegaly, and 40 bilateral hydrocele”. Has the cardiomegaly, ascites and pleurisy fully recovered? If so, during case presentation you should emphasize this aspect.
  18. section 3 should be named “discussion and conclusion” or you should make a new section for ”conclusion”
  19. The conclusion presented in the end of the manuscript somehow differs too much from the abstract conclusion. Maybe you can fix it.

Reviewer 2 Report

This is a well-written case report that should be of interest to obstetricians, neonatologists, hematologists, and transfusion medicine experts. However, there are several modifications, described below, that could enhance the manuscript.

  1. Of most importance, the authors must make a stronger case that more typical causes of severe intrauterine anemia and hydrops have been ruled-out. Was an antibody screen performed on the mother to rule-out alloimmunization? If so, this data (ABO/Rh, antibody screen) should be included. Similarly, the authors state that "The test for viral infection was negative...", but they should state exactly what viral testing was performed.
  2. Additional laboratory testing data for the neonate should be included in the manuscript including results of immunohematology testing (ABO/Rh, antibody screen, Direct Coombs' test). In addition, the authors state that the baby was pancytopenic, but they provide very little data to support this contention except for a single initial hemoglobin of 6.5 g/dL. They should include additional findings such as the white blood cell count and platelet count. Any additional diagnostic testing (e.g. for viral infectious disease) should also be mentioned.
  3. Was there any clinical or laboratory evidence of exocrine pancreas deficiency? If so, this too should be included in the manuscript.
  4. The case report makes it seem that a mitochondrial disease was suspected and diagnosed quickly. This hardly seems believable. It would be useful to provide more detail about the neonate's 2-month clinical course and the progression of clinical and diagnostic testing that made the authors focus on the possibility of a mitochondrial disorder.  

Round 2

Reviewer 1 Report

Thank you for your clarifications, and I believe that the changes you have made have increased the value of the manuscript. Congratulations on your work!

Author Response

I appreciate for your valuable suggestion.

Reviewer 2 Report

This is a fine revision, but I have 1 additional suggestion. The final sentence of the abstract should be deleted. It doesn't make much sense as written. The same idea is included in the Conclusion portion of the paper and is better explained there.

Author Response

Thank you very much for your valuable suggestion. We deleted the final sentence of the abstract.

We really appreciate your advice and critical reviews.
